# USE: Uncertainty Structure Estimation for Robust Semi-Supervised Learning

## Abstract

In this study, a novel idea, Uncertainty Structure Estimation (USE), a lightweight, algorithm-agnostic procedure that emphasizes the often-overlooked role of unlabeled data quality is introduced for Semi-supervised learning (SSL). SSL has achieved impressive progress, but its reliability in deployment is limited by the quality of the unlabeled pool. In practice, unlabeled data are almost always contaminated by out-of-distribution (OOD) samples, where both near-OOD and far-OOD can negatively affect performance in different ways. We argue that the bottleneck does not lie in algorithmic design, but rather in the absence of principled mechanisms to assess and curate the quality of unlabeled data. The proposed USE trains a proxy model on the labeled set to compute entropy scores for unlabeled samples, and then derives a threshold, via statistical comparison against a reference distribution, that separates informative (structured) from uninformative (structureless) samples. This enables assessment as a preprocessing step, removing uninformative or harmful unlabeled data before SSL training begins. Through extensive experiments on imaging (CIFAR-100) and NLP (Yelp Review) data, it is evident that USE consistently improves accuracy and robustness under varying levels of OOD contamination. Thus, it can be concluded that the proposed approach reframes unlabeled data quality control as a structural assessment problem, and considers it as a necessary component for reliable and efficient SSL in realistic mixed-distribution environments.

## 1 Introduction

Semi-supervised learning (SSL) has emerged as a core paradigm for reducing the reliance on large-scale labeled datasets in modern machine learning. By leveraging a large pool of unlabeled data alongside a relatively small labeled dataset, SSL methods have achieved remarkable success across computer vision, natural language processing, and audio processing (Wang et al. (2022)). However, a critical gap remains between benchmark settings and real-world scenarios: existing SSL methods assume that unlabeled data are drawn from the same distribution as the labeled data. In practice, unlabeled datasets are almost always contaminated by out-of-distribution (OOD) samples (Saito et al. (2021)), corrupted signals (Carmon et al. (2019)), or noisy labels (Li et al. (2020)). Such contamination can sharply degrade performance, not because the algorithm itself is flawed, but because it is trained on data lacking meaningful structure. To address this issue, existing SSL methods introduce increasingly complex algorithmic remedies, such as pseudo-label sharpening (Lee et al. (2013)), sample reweighting (Ren et al. (2018)), or consistency regularization (Laine & Aila (2017)), to mitigate these effects, a complementary direction is to place greater emphasis on assessing and improving the quality of unlabeled data itself, which may provide a simpler and more general path toward robust SSL.

Among different types of contamination, OOD samples are especially detrimental (Zhao et al. (2022)). We distinguish between *near-OOD* samples, which lie close to the in-distribution (ID) manifold and confuse decision boundaries, and *far-OOD* samples, which are unrelated to the task and induce nearly uniform predictive probabilities (Yang et al. (2024)). Both types can degrade SSL in distinct ways, yet their effects are systematically revealed in *entropy space*: ID samples cluster at low entropy, near-OOD samples approximate a uniform profile, and far-OOD samples concentrate at high entropy, as shown in Fig. 1. This observation motivates a broader perspective: the central

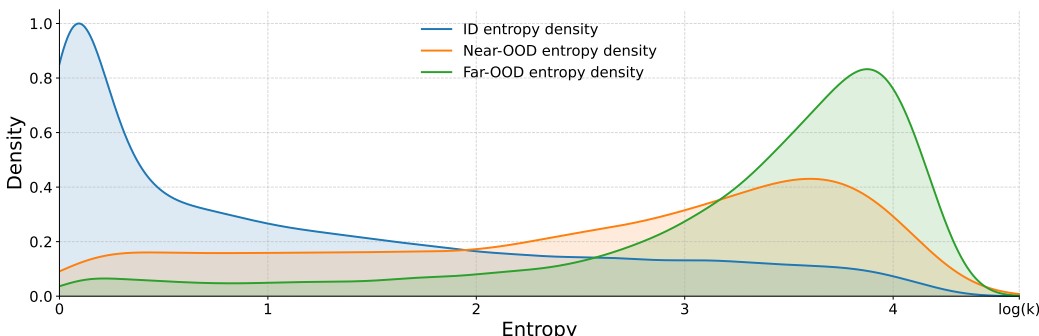

Figure 1: This figure displays the entropy density distribution computed using a proxy model trained on 200 labeled samples from the CIFAR-100 dataset. The blue curve represents in-distribution (ID) samples (CIFAR-100), concentrated in the low-entropy region; the orange curve represents near-OOD samples (Tiny ImageNet), exhibiting an approximately uniform distribution; and the green curve represents far-OOD samples (SVHN), concentrated in the high-entropy region.

problem is not simply detecting OOD samples one by one, but quantifying the *structural quality* of the unlabeled pool as a whole.

Building on this insight, we introduce **USE (Uncertainty Structure Estimation)**, a lightweight and algorithm-agnostic procedure for quantifying the *structural quality* of unlabeled data. USE trains a proxy model on the labeled set, computes entropy scores for all unlabeled samples, and determines a threshold, derived from statistical comparison against a reference distribution, that separates informative (structured) samples from uninformative (structureless) ones. Unlike traditional OOD filtering, which classifies individual samples, USE directly operationalizes distribution-level structural assessment as a preprocessing step, discarding harmful unlabeled data before SSL training begins.

**Our Contributions.** **(i)** We emphasize the importance of unlabeled data quality in SSL and introduce USE (Uncertainty Structure Estimation) as a principled entropy-based structural quality measure. **(ii)** We show that USE provides a lightweight and algorithm-agnostic procedure for unlabeled data quality control, in contrast to heuristic OOD filtering methods. **(iii)** Through extensive experiments on CIFAR-100 (200 and 1000 labeled samples) and Yelp Review, we demonstrate that USE consistently improves accuracy and robustness under varying levels of OOD contamination.

Together, these results reframe unlabeled data quality control as a structural assessment problem, establishing it as a necessary component for reliable and efficient SSL in realistic mixed-distribution environments.

## 2 RELATED WORK

**Semi-Supervised Learning.** Early SSL methods centered on two pillars: *pseudo-labeling*, which assigns artificial labels to unlabeled data based on model predictions (Lee et al. (2013)), and *consistency regularization*, which enforces prediction stability under input perturbations (Laine & Aila (2017)). Subsequent developments strengthened these ideas through strong data augmentations and distribution alignment, leading to methods such as MixMatch (Berthelot et al. (2019)), UDA (Xie et al. (2020)), FixMatch (Sohn et al. (2020)), and FlexMatch (Zhang et al. (2021)). A key trend in this later line of work is the use of *confidence-based filtering*, where only unlabeled examples with high-confidence predictions are included in training. While this mechanism works effectively in clean or mildly contaminated settings, it often breaks down when the unlabeled pool is heavily contaminated with OOD data or unseen classes. In such cases, sample-level heuristics or confidence filtering can fail, as OOD samples may still yield high-confidence predictions or distort decision boundaries. Our work differs in scope: rather than relying on confidence thresholds within the learning algorithm, we assess the *structural quality* of the unlabeled pool as a whole prior to training.

**OOD Detection.** OOD detection aims to distinguish in-distribution (ID) samples from those outside the training label space, and has been extensively studied in both post-hoc and training-time settings. Post-hoc methods apply scoring functions such as maximum softmax probability (MSP) (Hendrycks & Gimpel (2017)), ODIN (Liang et al. (2018)), or energy-based scores (Liu et al. (2020)), while training-time approaches include outlier exposure and logit regularization. Recent benchmarks such as OpenOOD v1.5 (Zhang et al. (2023)) emphasize the difficulty of detecting both near-OOD and far-OOD samples under realistic conditions. However, these approaches focus on per-sample classification and often require calibration, architectural changes, or additional supervision, which limits their applicability to SSL. Our method is different in scope: we do not treat OOD detection as a separate task, but instead introduce an entropy-based measure of *structural quality* that serves as a lightweight, algorithm-agnostic procedure for SSL, removing samples that lack structure, while preserving those more likely to carry task-relevant information.

**Evaluation Protocols.** Reliable evaluation requires standardized benchmarks that can disentangle algorithmic progress from artifacts of experimental design. To ensure consistency across SSL baselines, we adopt USB: A Unified Semi-supervised Learning Benchmark for Classification (Wang et al. (2022)), which provides unified datasets, splits, and training protocols for fair comparison of SSL methods. To characterize the effect of OOD contamination, we further build on RE-SSL (He et al. (2025)), which rigorously controls the ratio of seen-class and unseen-class samples in the unlabeled pool and introduces multiple robustness metrics to assess SSL performance under varying OOD levels. Finally, for our computer vision experiments, we leverage OpenOOD v1.5 (Zhang et al. (2023)), an enhanced benchmark for OOD detection that carefully curates ID and OOD datasets, including both near-OOD and far-OOD groups, while eliminating any category overlap between labeled and OOD sets.

## 3 METHODOLOGY

Our objective is to improve the reliability of semi-supervised learning (SSL) by explicitly assessing the *structural quality* of unlabeled data before training. Unlike traditional OOD detection methods, which attempt to classify individual samples, our approach introduces a *distribution-level assessment* that measures to what extent the unlabeled set as a whole carries task-relevant structure versus structureless noise. To this end, we propose **USE (Uncertainty Structure Estimation)**, a lightweight and algorithm-agnostic procedure grounded in entropy statistics. USE defines a principled entropy threshold that separates informative from uninformative samples in a single preprocessing step, reshaping the unlabeled pool while leaving downstream SSL algorithms unchanged.

### 3.1 PROBLEM SETUP

We consider a $k$-class classification problem in SSL. Let $\mathcal{L}$ denote the labeled dataset and $\mathcal{U}$ the unlabeled pool, which may contain a mixture of in-distribution (ID) and out-of-distribution (OOD) samples. We train a proxy model $f_\theta$ on $\mathcal{L}$ only, and for each $x \in \mathcal{U}$ compute the predictive distribution $p(c \mid x)$ over $k$ classes. The uncertainty of each prediction is measured by Shannon entropy (Shannon (1948)):

$$h(x) = -\sum_{c=1}^{k} p(c \mid x) \log p(c \mid x). \tag{1}$$

The entropy score $u = h(x)$ lies in $[0, \log k]$. Collecting $\{u_i\}_{i=1}^{n}$ over $\mathcal{U}$ yields an empirical distribution of uncertainty values. Our goal is to define a principled threshold $u^*$ that separates structured (informative) samples from structureless (uninformative or OOD) samples.

### 3.2 FROM DENSITY TO CDF: LINKING TO KS INTUITION

To estimate the entropy distribution, we apply kernel density estimation (KDE) (Parzen (1962)):

$$\hat{p}(u) = \frac{1}{nh} \sum_{i=1}^{n} K\left(\frac{u - u_i}{h}\right), \quad u \in [0, \log k], \tag{2}$$

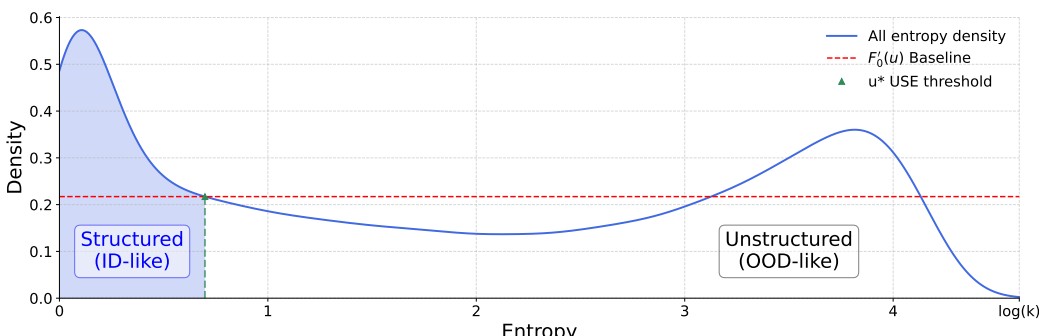

Figure 2: Illustration of the USE threshold. The threshold $u^*$ corresponds to the first descending downward crossing of the entropy density, separating informative low-entropy samples from uninformative high-entropy ones. The intersection with the reference curve density $F_0'(u)$ can be interpreted as the point of maximum structural discrepancy.

with bandwidth $h > 0$ and symmetric kernel $K(\cdot)$. The corresponding cumulative distribution function (CDF) is

$$\hat{F}(u) = \int_0^u \hat{p}(t)\, dt. \tag{3}$$

We interpret this through the intuition of the Kolmogorov-Smirnov (KS) test (Kolmogorov (1933)): if the unlabeled pool carried no structure, entropy scores would follow a reference distribution representing pure noise. Deviations between $\hat{F}(u)$ and such a reference reveal the degree of structure present. While we do not conduct a formal hypothesis test, this perspective provides a principled justification for defining thresholds based on entropy distributional differences.

$$D = \max_u \left| \hat{F}(u) - F_0(u) \right|, \tag{4}$$

where $\hat{F}(u)$ is the empirical cumulative distribution function (CDF) of the entropy scores, and $F_0(u)$ is the CDF of a chosen reference distribution. Intuitively, $\hat{F}(u)$ captures how uncertainty values are actually distributed in the unlabeled pool, while $F_0(u)$ provides a reference curve describing how the values would behave under a structureless assumption. This distributional discrepancy $D$ motivates our threshold as the point of maximum deviation in density space, rather than a direct statistical test. The discrepancy $D$ then quantifies how strongly the observed pool deviates from the reference, and serves as the basis for defining thresholds in USE.

### 3.3 THRESHOLD DEFINITION

The USE threshold $u^*$ is defined as the point where the empirical entropy distribution transitions from structured (informative) to structureless (uninformative) behavior. Specifically, it corresponds to the first downward intersection point between the empirical entropy density $\hat{p}(u)$ and the reference curve density $F_0'(u)$. Intuitively, this is the location where the discrepancy between the empirical distribution and the structureless reference is maximized, marking the onset of high-entropy regions dominated by OOD or noisy samples (Figure 2).

We define the USE threshold $u^*$ as the first downward intersection between $\hat{p}(u)$ and $F_0'(u)$:

$$u^* = \min \left\{ u \in (0, \log k) \,\middle|\, \hat{p}(u) = F_0'(u),\ \tfrac{d}{du}\hat{p}(u) \le 0 \right\}. \tag{5}$$

This choice satisfies desirable properties: it reflects the transition from ID-like, low-entropy concentration (due to proxy training) to structureless high-entropy regions, remains consistent even under severe OOD contamination, and avoids hand-tuned thresholds.

As shown in Fig. 2, the USE threshold corresponds to the first downward intersection between the empirical density and the reference curve $F_0'(u)$, marking the transition from structured to unstructured entropy regions.

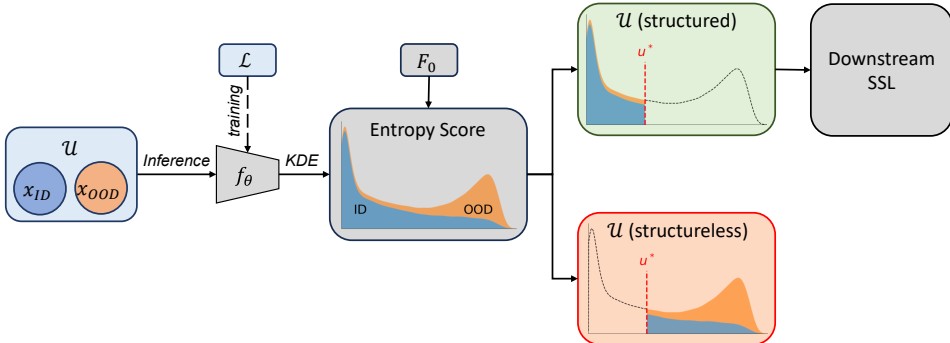

Figure 3: Overview of the USE pipeline. A proxy model trained on labeled data computes entropy scores for the unlabeled pool. By comparing the empirical entropy distribution with a structureless reference curve, a threshold $u^*$ is determined to separate structured from structureless samples, ensuring that downstream SSL is trained only on data with meaningful structure.

### 3.4 INTEGRATION INTO SSL FRAMEWORKS

USE is applied as a lightweight, algorithm-agnostic procedure: entropy scores are computed from the proxy model, and samples with $u > u^*$ are discarded before downstream SSL training. Figure 3 summarizes the overall procedure.

## 4 EXPERIMENTAL SETUP

This section details the datasets, baseline algorithms, evaluation metrics, and implementation protocols. We follow the USB benchmark framework (Wang et al. (2022)) to ensure comparability across domains and methods. Our USE is applied as a plug-in stage prior to downstream SSL training, without altering existing algorithms. To assess robustness under OOD contamination, we report both standard classification accuracy and RE-SSL robustness indicators (He et al. (2025)).

### 4.1 DATASETS

**Computer Vision (CV).** We evaluate on CIFAR-100 (Krizhevsky et al. (2009)) as the in-distribution (ID) dataset, with two label budgets: (i) 200 labeled samples (2 per class) and (ii) 1,000 labeled samples (10 per class). The unlabeled pool contains 50,000 ID samples, contaminated by Tiny ImageNet (near-OOD) (Le & Yang (2015)) and SVHN (far-OOD) (Netzer et al. (2011)). Standard augmentations (crop, flip, and color jitter) are applied. When OOD samples are insufficient to meet the target ratio, we oversample them using standard augmentations.

**Natural Language Processing (NLP).** For NLP experiments, we adopt Yelp Review (Asghar (2016)) as the ID dataset, sampling 250 labeled examples (50 per class). The unlabeled pool is contaminated with IMDB (near-OOD) (Maas et al. (2011)) and AGNews (far-OOD) (Zhang et al. (2015)). Augmentation follows the EDA (Wei & Zou (2019)) pipeline with default hyperparameters.

**Contamination ratio.** We define the OOD contamination ratio as

$$r = \frac{D_{\text{OOD}}}{D_{\text{ID}} + D_{\text{OOD}}}, \tag{6}$$

where $D_{\text{ID}}$ and $D_{\text{OOD}}$ denote the counts of ID and OOD unlabeled samples. For CV: $r \in \{0.0, 0.2, 0.4, 0.5, 0.6, 0.8\}$; for NLP: $r \in \{0.0, 0.2, 0.4, 0.6, 0.8\}$. Following RE-SSL (He et al. (2025)), the number of ID unlabeled samples is fixed while only the number of OOD samples is varied, thereby isolating contamination impact on robustness metrics.

## 4.2 SSL BASELINES

We include representative SSL methods: Pseudo-Label (Lee et al. (2013)), VAT (Miyato et al. (2018)), MixMatch (Berthelot et al. (2019)), UDA (Xie et al. (2020)), FixMatch (Sohn et al. (2020)), and FlexMatch (Zhang et al. (2021)). We follow USB (Wang et al. (2022)) for broad coverage across vision and language tasks, testing the algorithm-agnostic nature of USE. For NLP experiments, Mix-Match is excluded. As noted in USB, sequence-level mixing is problematic: concatenating or interpolating text of varying lengths often produces degenerate sequences and harms performance (Wang et al. (2022)). VAT does not run natively for NLP data without modality-aware modifications. We use RoBERTa (Liu et al. (2019)) as the backbone for NLP tasks, which shares the same architecture as BERT but generally offers stronger downstream performance. We restrict NLP baselines to four widely used methods: Pseudo-Label, UDA, FixMatch, and FlexMatch.

## 4.3 REFERENCE CURVE

A central step in USE is to compare the empirical CDF of entropy values against a reference distribution in order to detect deviations that signal structural quality. In all our experiments, we adopt the **uniform distribution on the entropy axis**, i.e., a flat density over $u \in [0, \log k]$ rather than a uniform class-probability assumption for individual samples. Formally,

$$F_0(u) = \frac{u}{\log k}, \qquad F_0'(u) = \frac{1}{\log k}, \quad u \in [0, \log k]. \tag{7}$$

This serves as a simple *distribution-level null hypothesis*: it represents an unlabeled pool where entropy values are spread evenly across the full support, with no systematic concentration in either low- or high-uncertainty regions. Among distributions on $[0, \log k]$, the uniform law maximizes Shannon entropy, serving as the least-informative prior.

Importantly, the choice of reference curve is modular and not tied to the method. Other priors can be substituted to reflect different assumptions or domains. We leave a systematic exploration of such alternatives to future work.

## 4.4 METRICS

**Standard metrics.** We report top-1 classification accuracy, defined as the proportion of test samples for which the predicted label with the highest probability matches the ground-truth class. The test set is strictly ID and contains no OOD samples, ensuring that reported accuracy reflects performance on the target classification task rather than robustness to distributional shifts.

**Robustness metrics (RE-SSL).** To quantify robustness beyond accuracy, we adopt the indicators defined in RE-SSL (He et al. (2025)):

- **Rslope**: regression slope of accuracy vs. $r$, capturing global robustness.
- **GM**: global deviation from mean performance, reflecting sensitivity to contamination.
- **WAD / BAD**: worst / best adjacent drops between successive $r$ values, capturing local robustness.
- $P_{AD \geq 0}$: proportion of adjacent intervals where accuracy does not decrease.

These metrics jointly characterize both global and local robustness, complementing conventional evaluation.

## 4.5 IMPLEMENTATION DETAILS

All experiments are conducted within the USB framework using identical training protocols for fair comparison. For CV tasks we adopt ViT backbones, while for NLP we employ BERT-based classifiers. Models are trained on NVIDIA A100/V100 GPUs. Our proxy model trains on labeled data, computes entropy scores for all unlabeled samples, and discards those above the USE threshold. This procedure introduces negligible computational overhead (e.g., $\sim 5\%$ extra time on CIFAR-100 with 250 labels), and leaves the downstream SSL pipeline unchanged, thereby fulfilling the principle of a lightweight and plug-and-play procedure.

Table 1: Mean top-1 accuracy over OOD contamination ratios ($r = \frac{D_{\text{OOD}}}{D_{\text{ID}}+D_{\text{OOD}}}$) across datasets.

| | | CIFAR-100 (200 labeled) | | CIFAR-100 (1000 labeled) | | Yelp (250 labeled) | |
|---|---|---|---|---|---|---|---|
| Supervised | | 0.6469 | | 0.8010 | | 0.5992 | |
| OOD types | | Tiny ImageNet | SVHN | Tiny ImageNet | SVHN | IMDB | AGNews |
| Pseudo Label | Base | 0.6644 | 0.6763 | 0.8047 | 0.8122 | 0.5968 | 0.5978 |
| | + USE | 0.6741 | 0.6778 | 0.8120 | 0.8168 | 0.5971 | 0.5971 |
| FixMatch | Base | 0.6570 | 0.6780 | 0.8270 | 0.8380 | 0.6187 | 0.6162 |
| | + USE | 0.6696 | 0.6693 | 0.8446 | 0.8504 | 0.6201 | 0.6233 |
| FlexMatch | Base | 0.7010 | 0.6983 | 0.8222 | 0.8354 | 0.6206 | 0.6230 |
| | + USE | 0.7041 | 0.6961 | 0.8404 | 0.8473 | 0.6223 | 0.6219 |
| UDA | Base | 0.6915 | 0.7157 | 0.8288 | 0.8372 | 0.6180 | 0.6175 |
| | + USE | 0.6971 | 0.7031 | 0.8446 | 0.8517 | 0.6185 | 0.6231 |
| MixMatch | Base | 0.6260 | 0.5425 | 0.7999 | 0.7898 | N/A | N/A |
| | + USE | 0.6611 | 0.6595 | 0.8049 | 0.8015 | | |
| VAT | Base | 0.6179 | 0.7034 | 0.8103 | 0.8330 | N/A | N/A |
| | + USE | 0.7148 | 0.7194 | 0.8370 | 0.8438 | | |

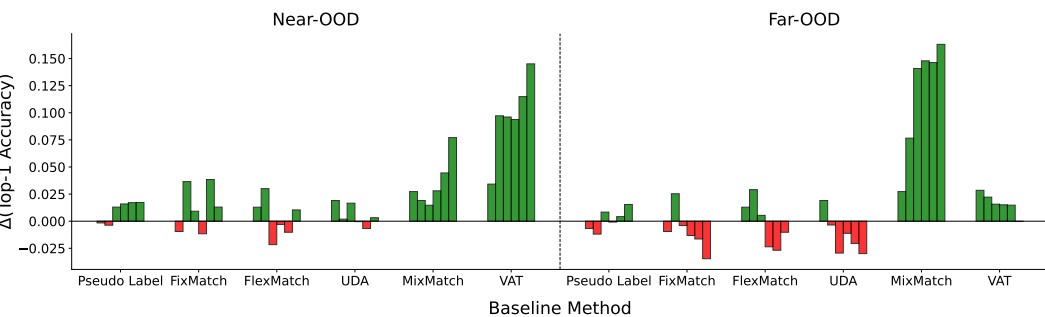

Figure 4: Performance comparison of SSL methods on CIFAR-100 with 200 labeled samples. For each method, the six bars show the top-1 accuracy difference (USE – baseline) under contamination ratios $r \in \{0.0, 0.2, 0.4, 0.5, 0.6, 0.8\}$, with Tiny ImageNet (near-OOD) and SVHN (far-OOD) used as contamination sources. Green bars indicate that USE outperforms the baseline, while red bars indicate the opposite.

## 5 RESULTS

Table 1 summarizes average accuracy across contamination ratios $r$ for different datasets and label budgets, comparing SSL baselines with and without USE. In this section, we introduce the most challenging case of CIFAR-100 with 200 labels, then consider the 1,000-label setting where stronger proxies enhance USE's effect. We next evaluate Yelp Review to test another modality generality, and finally complement accuracy results with robustness metrics.

### 5.1 CIFAR-100 WITH 200 LABELS UNDER OOD CONTAMINATION

Figure 4 shows that USE yields consistent gains: most bars are green, while negative effects are rare and small. Table 1 confirms the trend, reporting higher average accuracy with USE across all methods.

**Near-OOD.** For near-OOD contamination (Tiny ImageNet), USE provides steady improvements. For example, FlexMatch rises from 0.7010 to 0.7041, UDA improves from 0.6915 to 0.6971, and VAT improves strongly from 0.6179 to 0.7148. The trend is clear: when the contamination ratio increases, baselines often decline, but USE stabilizes performance.

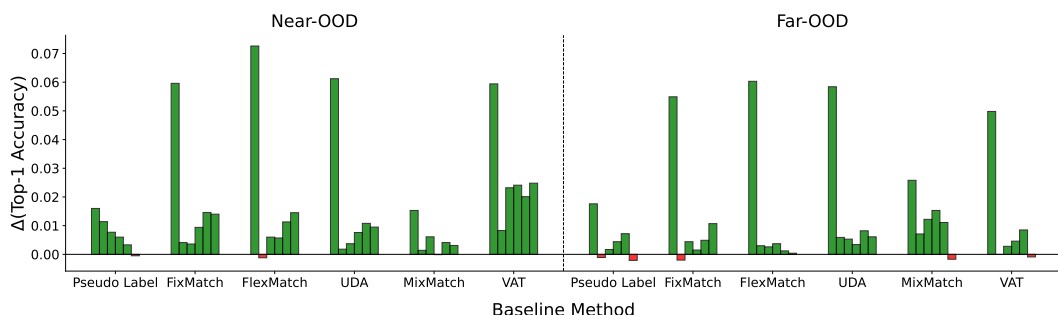

Figure 5: Performance comparison of SSL methods on CIFAR-100 with 1000 labeled samples. For each method, the six bars show the top-1 accuracy difference (USE - baseline) under contamination ratios $r \in \{0.0, 0.2, 0.4, 0.5, 0.6, 0.8\}$, with Tiny ImageNet (near-OOD) and SVHN (far-OOD) used as contamination sources. Green bars indicate that USE outperforms the baseline, while red bars indicate the opposite.

**Far-OOD.** With SVHN contamination, effects are mixed. Confidence-based methods (FixMatch, FlexMatch, UDA) include built-in mechanisms that mask low-confidence samples, so USE offers little additional benefit (e.g., FixMatch drops from 0.6780 to 0.6693). In contrast, methods without such masking gain substantially: MixMatch jumps from 0.5425 to 0.6595, and VAT from 0.7034 to 0.7194. This shows USE is especially effective when the base method cannot filter OOD data on its own.

**Conclusion.** Overall, USE improves SSL under OOD contamination, with positive effects much larger and more common than negative ones. The size of the gain depends on the method: confidence masks already handle far-OOD, but USE adds value for near-OOD and is critical for methods without masking.

### 5.2 PROXY MODEL SENSITIVITY: CIFAR-100 WITH 1000 LABELS

Figure 5 shows that with 1,000 labels, USE yields larger and more consistent gains than in the 200-label case, with red bars now rare and small. Table 1 confirms that every method achieves higher average accuracy with USE. These results highlight a scaling effect: as proxy quality improves, entropy estimates sharpen, enabling USE to more effectively separate structured from structureless samples and smooth performance under contamination.

### 5.3 EVALUATION ON YELP REVIEW (NLP DOMAIN)

Compared to vision benchmarks, OOD contamination on text classification has a weaker effect. Still, Table 1 shows that USE provides consistent improvements across most methods. For example, FixMatch improves from 0.6187 to 0.6233 under far-OOD, and UDA improves from 0.6180 to 0.6231. This indicates that USE generalizes beyond images to NLP tasks.

### 5.4 ROBUSTNESS ASSESSMENT VIA RE-SSL METRICS

We further assess robustness using the five metrics introduced in RE-SSL (He et al. (2025)): Rslope, GM, BAD, WAD, and $P_{AD \geq 0}$. These metrics capture both global stability (Rslope, GM), local fluctuations (BAD, WAD), and the proportion of positive transitions ($P_{AD \geq 0}$). Table 2 summarizes the impact of USE on each metric compared to the baseline.

**CIFAR-100 with 200 labels.** In the low-data setting, USE shows clear benefits on robustness. For far-OOD, four out of five metrics improve: Rslope is closer to zero, GM increases, BAD and WAD

Table 2: Robustness comparison (USE vs. w/o). ↑ better, ↓ worse, → neutral.

| Setting | Rslope | GM | BAD | WAD | $P_{AD \geq 0}$ |
|---|---|---|---|---|---|
| C100/200/Far | ↑ | ↑ | ↑ | ↑ | ↓ |
| C100/200/Near | ↑ | ↑ | ↓ | ↑ | → |
| C100/1k/Far | ↓ | ↑ | ↑ | ↓ | ↓ |
| C100/1k/Near | ↓ | ↑ | ↑ | ↑ | ↓ |
| Yelp/250/Near | ↑ | ↑ | ↑ | ↓ | ↑ |
| Yelp/250/Far | ↑ | ↓ | ↓ | ↑ | ↓ |

become smaller, and only $P_{AD \geq 0}$ drops slightly. For near-OOD, Rslope, GM, and WAD all improve, while $P_{AD \geq 0}$ remains unchanged and BAD becomes worse. This indicates that USE not only raises accuracy (as shown in Sec. 5.1), but also smooths the performance curve across contamination ratios, making models more reliable in both global and local perspectives.

**CIFAR-100 with 1000 labels.**    With more labels, the robustness picture is more mixed. USE consistently increases GM, meaning the worst-case accuracy across contamination ratios is better preserved. However, Rslope becomes more negative, suggesting that the overall decline with increasing contamination is sharper. Similarly, $P_{AD \geq 0}$ tends to decrease, indicating fewer contamination increments lead to non-decreasing performance. For local stability, results vary: WAD improves in the near-OOD case but slightly worsens in far-OOD, while BAD consistently decreases, showing that large performance jumps are suppressed. Together, these results suggest that when the proxy is stronger, USE primarily enhances the worst-case guarantee (GM) and reduces local volatility (BAD), but may also introduce a steeper overall decline (Rslope). This reflects the sensitivity of USE to proxy quality: a stronger proxy gives sharper entropy estimates, which both helps filter structureless samples and accentuates the tradeoff between global slope and local stability.

**Yelp Review with 250 labels.**    In NLP tasks, the robustness effect of USE is less consistent. For far-OOD, Rslope and WAD both improve, showing smoother overall trends and fewer local fluctuations, but GM, BAD, and $P_{AD \geq 0}$ decrease slightly. The drops are very small, further confirming that the instability mainly comes from sampling noise rather than systematic failure. Overall, this reflects the intrinsic variability of NLP benchmarks: OOD contamination is less impactful than in vision, and consequently, the corrective effect of USE is weaker and less stable. This aligns with our earlier observation that while USE is broadly helpful, its benefits in text classification are more modest and sensitive to data scale.

**Comparison across settings.**    The contrast between 200 and 1000 labels reveals a key trend. With fewer labels, USE acts as a broad stabilizer, improving most robustness scores simultaneously. With more labels, the improvements concentrate on GM and BAD, while Rslope and $P_{AD \geq 0}$ show regression. This highlights that the utility of USE is not uniform: it depends on the interplay between proxy strength and the contamination type. In practice, this means USE is most valuable in extremely low-label setting, where robustness improvements are broad and consistent.

**Summary.**    USE enhances robustness across settings: while some metrics regress, GM and BAD improve reliably, and stability is preserved. This shows that USE strengthens SSL against OOD contamination without harming overall robustness. Full details appear in the appendix.

## 6    DISCUSSION AND LIMITATIONS

**Discussion.**    USE aims to highlight unlabeled data quality as a core issue in SSL. While most research advances focus on algorithm design, USE provides a simple way to filter structureless samples before training. It is complementary to existing SSL methods and flexible through the choice of $F_0$.

**Limitations.**    USE relies purely on entropy, which may be insufficient for capturing more complex data structures, and our current evaluation is restricted to classification tasks.

**Future Work.**    A promising direction is to incorporate richer uncertainty signals (e.g., energy-based or contrastive scores) and to extend the approach to multimodal and generative settings.

## 7    CONCLUSION

We introduced USE, a lightweight and algorithm-agnostic procedure for assessing unlabeled data in SSL. By filtering out structureless samples, USE improves accuracy and robustness across vision and NLP benchmarks, especially in low-label settings. This work reframes unlabeled data quality as a key factor in SSL and offers a complementary direction alongside algorithmic innovations.

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

## A  APPENDIX

### A.1  DETAILED RESULTS ON CIFAR-100 WITH 200 LABELS

Table 3 reports the detailed results on CIFAR-100 with 200 labeled samples under varying contamination ratios $r \in \{0.0, 0.2, 0.4, 0.5, 0.6, 0.8\}$. We evaluate both near-OOD (Tiny ImageNet) and far-OOD (SVHN) settings across representative SSL baselines, with and without the proposed USE procedure. In addition to standard accuracy, we include robustness indicators defined in RE-SSL: **Rslope** (slope of accuracy vs. contamination), **GM** (global deviation from mean performance), **BAD/WAD** (best/worst adjacent drops), and $P_{AD \geq 0}$ (proportion of non-decreasing intervals).

Table 3: CIFAR-100 with 200 labeled samples.

| Method | r=0.0 | r=0.2 | r=0.4 | r=0.5 | r=0.6 | r=0.8 | Avg. | Rslope | GM | BAD | WAD | $P_{AD\geq0}$ |
|---|---|---|---|---|---|---|---|---|---|---|---|---|
| Supervised | 0.6469 | | | | | | | | | | | |
| near-OOD (Tiny ImageNet) | | | | | | | | | | | | |
| Pseudo Label | 0.6757 | 0.6776 | 0.6604 | 0.6591 | 0.6571 | 0.6565 | 0.6644 | -0.0301 | 0.0490 | 0.0095 | -0.0860 | 0.2000 |
| Pseudo Label + USE | 0.6741 | 0.6739 | 0.6734 | 0.6750 | 0.6742 | 0.6738 | 0.6741 | 0.0001 | 0.0022 | 0.0160 | -0.0080 | 0.2000 |
| FixMatch | 0.6752 | 0.6522 | 0.6691 | 0.6692 | 0.6357 | 0.6403 | 0.6570 | -0.0393 | 0.0853 | 0.0845 | -0.3350 | 0.6000 |
| FixMatch + USE | 0.6657 | 0.6887 | 0.6783 | 0.6576 | 0.6741 | 0.6533 | 0.6696 | -0.0222 | 0.0645 | 0.1650 | -0.2070 | 0.4000 |
| FlexMatch | 0.7076 | 0.7077 | 0.7161 | 0.7025 | 0.7017 | 0.6702 | 0.7010 | -0.0392 | 0.0615 | 0.0420 | -0.1575 | 0.4000 |
| FlexMatch + USE | 0.7205 | 0.7377 | 0.6945 | 0.6995 | 0.6915 | 0.6806 | 0.7041 | -0.0628 | 0.1002 | 0.0860 | -0.2160 | 0.4000 |
| UDA | 0.7128 | 0.7033 | 0.6910 | 0.6946 | 0.6866 | 0.6608 | 0.6915 | -0.0584 | 0.0723 | 0.0360 | -0.1290 | 0.2000 |
| UDA + USE | 0.7319 | 0.7052 | 0.7076 | 0.6942 | 0.6799 | 0.6640 | 0.6971 | -0.0796 | 0.1066 | 0.0120 | -0.1430 | 0.2000 |
| MixMatch | 0.6342 | 0.6401 | 0.6418 | 0.6348 | 0.6180 | 0.5870 | 0.6260 | -0.0549 | 0.0939 | 0.0295 | -0.1680 | 0.4000 |
| MixMatch + USE | 0.6615 | 0.6592 | 0.6565 | 0.6628 | 0.6625 | 0.6641 | 0.6611 | 0.0046 | 0.0130 | 0.0630 | -0.0135 | 0.4000 |
| VAT | 0.6976 | 0.6226 | 0.6180 | 0.6121 | 0.5985 | 0.5584 | 0.6179 | -0.1496 | 0.1692 | -0.0230 | -0.3750 | 0.0000 |
| VAT + USE | 0.7318 | 0.7198 | 0.7141 | 0.7059 | 0.7134 | 0.7035 | 0.7148 | -0.0330 | 0.0442 | 0.0750 | -0.0820 | 0.2000 |
| far-OOD (SVHN) | | | | | | | | | | | | |
| Pseudo Label | 0.6808 | 0.6862 | 0.6712 | 0.6786 | 0.6764 | 0.6648 | 0.6763 | -0.0199 | 0.0333 | 0.0740 | -0.0750 | 0.4000 |
| Pseudo Label + USE | 0.6741 | 0.6743 | 0.6796 | 0.6778 | 0.6806 | 0.6802 | 0.6778 | 0.0091 | 0.0143 | 0.0280 | -0.0180 | 0.6000 |
| FixMatch | 0.6752 | 0.6685 | 0.6721 | 0.6769 | 0.6791 | 0.6962 | 0.6780 | 0.0255 | 0.0386 | 0.0855 | -0.0335 | 0.8000 |
| FixMatch + USE | 0.6657 | 0.6938 | 0.6681 | 0.6637 | 0.6627 | 0.6616 | 0.6693 | -0.0206 | 0.0491 | 0.1405 | -0.1285 | 0.2000 |
| FlexMatch | 0.7076 | 0.6902 | 0.6890 | 0.7105 | 0.7118 | 0.6806 | 0.6983 | -0.0129 | 0.0701 | 0.2150 | -0.1560 | 0.4000 |
| FlexMatch + USE | 0.7205 | 0.7193 | 0.6944 | 0.6869 | 0.6850 | 0.6704 | 0.6961 | -0.0681 | 0.0953 | -0.0060 | -0.1245 | 0.0000 |
| UDA | 0.7128 | 0.7150 | 0.7303 | 0.7169 | 0.7109 | 0.7081 | 0.7157 | -0.0063 | 0.0317 | 0.0765 | -0.1340 | 0.4000 |
| UDA + USE | 0.7319 | 0.7115 | 0.7009 | 0.7057 | 0.6904 | 0.6782 | 0.7031 | -0.0623 | 0.0796 | 0.0480 | -0.1530 | 0.2000 |
| MixMatch | 0.6342 | 0.5833 | 0.5218 | 0.5130 | 0.5120 | 0.4906 | 0.5425 | -0.1828 | 0.2651 | -0.0100 | -0.3075 | 0.0000 |
| MixMatch + USE | 0.6615 | 0.6599 | 0.6626 | 0.6609 | 0.6583 | 0.6537 | 0.6595 | -0.0081 | 0.0139 | 0.0135 | -0.0260 | 0.2000 |
| VAT | 0.7002 | 0.7027 | 0.7054 | 0.7042 | 0.7016 | 0.7063 | 0.7034 | 0.0056 | 0.0114 | 0.0235 | -0.0260 | 0.6000 |
| VAT + USE | 0.7287 | 0.7249 | 0.7211 | 0.7193 | 0.7164 | 0.7060 | 0.7194 | -0.0264 | 0.0330 | -0.0180 | -0.0520 | 0.0000 |

**Method-specific behavior under OOD.** Pseudo Label and MixMatch degrade substantially with increasing contamination, especially in far-OOD where accuracy drops sharply, reflected in negative Rslope values and high BAD drops. Their reliance on raw pseudo-labels makes them vulnerable to noisy OOD samples. FixMatch, FlexMatch and UDA exhibit comparatively stable performance, as their confidence-based masking helps filter out uncertain samples. VAT shows mixed results, with significant drops in near-OOD but more resilience in far-OOD. VAT's adversarial perturbations can sometimes help navigate OOD noise, but its effectiveness varies with the contamination type.

**Impact of USE.** USE consistently improves average accuracy and robustness across weak baselines. For instance, Pseudo Label and MixMatch, USE nearly eliminates the negative slope and reduces volatility. yielding balanced Rslope (close to zero) and lower BAD/WAD values. Even for stronger baselines like FixMatch, FlexMatch, and UDA, USE provides modest but consistent gains in average accuracy and robustness metrics. The improvements are more pronounced in near-OOD settings, where the structural filtering by USE effectively mitigates the impact of ambiguous samples. In far-OOD scenarios, the benefits are more variable, as confidence-based methods already incorporate some filtering.

Overall, with only 200 labeled samples, USE serves as a decisive factor: it stabilizes fragile methods and reduces volatility even for strong baselines, improving RESSL robustness metrics (GM, BAD, WAD, and $P_{AD\geq0}$) across both near- and far-OOD settings.

## A.2 DETAILED RESULTS ON CIFAR-100 WITH 1000 LABELS

Table 4 presents the detailed results on CIFAR-100 with 1000 labeled samples. Compared to the 200-label case, the results show stronger and more consistent improvements from USE across all methods and contamination ratios. Average accuracy increases for every method when USE is applied, confirming its effectiveness with a stronger proxy model.

This comparison highlights that label availability fundamentally alters robustness profiles: at 200 labels, USE is a rescue mechanism; at 1000 labels, it is a stability enhancer.

## A.3 DETAILED RESULTS ON YELP REVIEW WITH 250 LABELS

Table 5 reports the results on the Yelp Review dataset with 250 labeled samples, under contamination ratios $r \in \{0.0, 0.2, 0.4, 0.6, 0.8\}$. Several observations emerge.

Table 4: CIFAR-100 with 1000 labeled samples.

| Method | r=0.0 | r=0.2 | r=0.4 | r=0.5 | r=0.6 | r=0.8 | Avg. | Rslope | GM | BAD | WAD | $P_{AD\geq0}$ |
|---|---|---|---|---|---|---|---|---|---|---|---|---|
| Supervised | 0.6469 | | | | | | | | | | | |
| near-OOD (Tiny ImageNet) | | | | | | | | | | | | |
| Pseudo Label | 0.8027 | 0.8111 | 0.8050 | 0.8032 | 0.8029 | 0.8031 | 0.8047 | -0.0040 | 0.0135 | 0.0420 | -0.0305 | 0.4000 |
| Pseudo Label + USE | 0.8187 | 0.8225 | 0.8127 | 0.8092 | 0.8062 | 0.8026 | 0.8120 | -0.0244 | 0.0359 | 0.0190 | -0.0490 | 0.2000 |
| FixMatch | 0.7940 | 0.8485 | 0.8428 | 0.8360 | 0.8289 | 0.8119 | 0.8270 | 0.0101 | 0.0963 | 0.2725 | -0.0850 | 0.2000 |
| FixMatch + USE | 0.8536 | 0.8526 | 0.8464 | 0.8454 | 0.8435 | 0.8259 | 0.8446 | -0.0314 | 0.0395 | -0.0050 | -0.0880 | 0.0000 |
| FlexMatch | 0.7875 | 0.8453 | 0.8383 | 0.8333 | 0.8241 | 0.8047 | 0.8222 | 0.0092 | 0.1044 | 0.2890 | -0.0970 | 0.2000 |
| FlexMatch + USE | 0.8601 | 0.8441 | 0.8443 | 0.8390 | 0.8354 | 0.8192 | 0.8404 | -0.0447 | 0.0549 | 0.0010 | -0.0810 | 0.2000 |
| UDA | 0.7952 | 0.8484 | 0.8465 | 0.8407 | 0.8304 | 0.8117 | 0.8288 | 0.0103 | 0.1015 | 0.2660 | -0.1030 | 0.2000 |
| UDA + USE | 0.8564 | 0.8502 | 0.8502 | 0.8483 | 0.8412 | 0.8212 | 0.8446 | -0.0380 | 0.0535 | 0.0000 | -0.1000 | 0.2000 |
| MixMatch | 0.7950 | 0.8068 | 0.7988 | 0.8014 | 0.7982 | 0.7994 | 0.7999 | 0.0005 | 0.0167 | 0.0590 | -0.0400 | 0.6000 |
| MixMatch + USE | 0.8103 | 0.8082 | 0.8049 | 0.8013 | 0.8023 | 0.8025 | 0.8049 | -0.0114 | 0.0173 | 0.0100 | -0.0360 | 0.4000 |
| VAT | 0.7878 | 0.8396 | 0.8180 | 0.8131 | 0.8110 | 0.7925 | 0.8103 | -0.0087 | 0.0807 | 0.2590 | -0.1080 | 0.2000 |
| VAT + USE | 0.8472 | 0.8479 | 0.8412 | 0.8372 | 0.8311 | 0.8173 | 0.8370 | -0.0375 | 0.0511 | 0.0035 | -0.0690 | 0.2000 |
| far-OOD (SVHN) | | | | | | | | | | | | |
| Pseudo Label | 0.8011 | 0.8159 | 0.8165 | 0.8184 | 0.8104 | 0.8106 | 0.8122 | 0.0081 | 0.0287 | 0.0740 | -0.0800 | 0.8000 |
| Pseudo Label + USE | 0.8187 | 0.8148 | 0.8182 | 0.8228 | 0.8176 | 0.8085 | 0.8168 | -0.0071 | 0.0205 | 0.0460 | -0.0520 | 0.4000 |
| FixMatch | 0.7987 | 0.8563 | 0.8492 | 0.8475 | 0.8444 | 0.8319 | 0.8380 | 0.0290 | 0.0908 | 0.2880 | -0.0625 | 0.2000 |
| FixMatch + USE | 0.8536 | 0.8543 | 0.8536 | 0.8490 | 0.8493 | 0.8426 | 0.8504 | -0.0136 | 0.0206 | 0.0035 | -0.0460 | 0.4000 |
| FlexMatch | 0.7998 | 0.8477 | 0.8439 | 0.8450 | 0.8421 | 0.8339 | 0.8354 | 0.0330 | 0.0742 | 0.2395 | -0.0410 | 0.4000 |
| FlexMatch + USE | 0.8601 | 0.8507 | 0.8465 | 0.8487 | 0.8433 | 0.8343 | 0.8473 | -0.0285 | 0.0354 | 0.0220 | -0.0540 | 0.2000 |
| UDA | 0.7980 | 0.8509 | 0.8508 | 0.8463 | 0.8424 | 0.8347 | 0.8372 | 0.0340 | 0.0833 | 0.2645 | -0.0450 | 0.2000 |
| UDA + USE | 0.8564 | 0.8568 | 0.8561 | 0.8497 | 0.8506 | 0.8408 | 0.8517 | -0.0188 | 0.0282 | 0.0090 | -0.0640 | 0.4000 |
| MixMatch | 0.7845 | 0.8025 | 0.7917 | 0.7874 | 0.7870 | 0.7859 | 0.7898 | -0.0068 | 0.0291 | 0.0900 | -0.0540 | 0.2000 |
| MixMatch + USE | 0.8103 | 0.8096 | 0.8039 | 0.8027 | 0.7981 | 0.7842 | 0.8015 | -0.0309 | 0.0413 | -0.0035 | -0.0695 | 0.0000 |
| VAT | 0.7983 | 0.8446 | 0.8416 | 0.8402 | 0.8384 | 0.8349 | 0.8330 | 0.0346 | 0.0694 | 0.2315 | -0.0180 | 0.2000 |
| VAT + USE | 0.8481 | 0.8446 | 0.8444 | 0.8448 | 0.8469 | 0.8340 | 0.8438 | -0.0124 | 0.0196 | 0.0210 | -0.0645 | 0.4000 |

Table 5: Yelp Review with 250 labeled samples.

| Method | r=0.0 | r=0.2 | r=0.4 | r=0.6 | r=0.8 | Avg. | Rslope | GM | BAD | WAD | $P_{AD\geq0}$ |
|---|---|---|---|---|---|---|---|---|---|---|---|
| Supervised | 0.5992 | | | | | | | | | | |
| near-OOD (IMDB) | | | | | | | | | | | |
| Pseudo Label | 0.5995 | 0.5985 | 0.5946 | 0.5966 | 0.5950 | 0.5975 | -0.0055 | 0.0086 | 0.0100 | -0.0195 | 0.2500 |
| Pseudo Label + USE | 0.5978 | 0.6015 | 0.5944 | 0.5935 | 0.5983 | 0.5971 | -0.0035 | 0.0126 | 0.0240 | -0.0355 | 0.5000 |
| FixMatch | 0.6230 | 0.6208 | 0.6176 | 0.6176 | 0.6146 | 0.6187 | -0.0100 | 0.0127 | 0.0000 | -0.0160 | 0.2500 |
| FixMatch + USE | 0.6228 | 0.6198 | 0.6202 | 0.6174 | 0.6204 | 0.6201 | -0.0036 | 0.0061 | 0.0150 | -0.0150 | 0.5000 |
| FlexMatch | 0.6251 | 0.6204 | 0.6195 | 0.6170 | 0.6212 | 0.6206 | -0.0056 | 0.0100 | 0.0210 | -0.0235 | 0.2500 |
| FlexMatch + USE | 0.6229 | 0.6237 | 0.6238 | 0.6215 | 0.6198 | 0.6223 | -0.0042 | 0.0068 | 0.0040 | -0.0115 | 0.5000 |
| UDA | 0.6205 | 0.6189 | 0.6166 | 0.6188 | 0.6150 | 0.6180 | -0.0055 | 0.0086 | 0.0110 | -0.0190 | 0.2500 |
| UDA + USE | 0.6218 | 0.6215 | 0.6204 | 0.6150 | 0.6136 | 0.6185 | -0.0115 | 0.0166 | -0.0015 | -0.0270 | 0.0000 |
| far-OOD (AGNews) | | | | | | | | | | | |
| Pseudo Label | 0.5987 | 0.5962 | 0.5995 | 0.5972 | 0.5976 | 0.5978 | -0.0006 | 0.0050 | 0.0165 | -0.0125 | 0.5000 |
| Pseudo Label + USE | 0.5957 | 0.5969 | 0.5978 | 0.5968 | 0.5982 | 0.5971 | 0.0024 | 0.0037 | 0.0070 | -0.0050 | 0.7500 |
| FixMatch | 0.6234 | 0.6148 | 0.6129 | 0.6139 | 0.6160 | 0.6162 | -0.0079 | 0.0144 | 0.0105 | -0.0430 | 0.5000 |
| FixMatch + USE | 0.6262 | 0.6195 | 0.6256 | 0.6223 | 0.6228 | 0.6233 | -0.0020 | 0.0105 | 0.0305 | -0.0335 | 0.5000 |
| FlexMatch | 0.6233 | 0.6234 | 0.6242 | 0.6219 | 0.6222 | 0.6230 | -0.0019 | 0.0038 | 0.0040 | -0.0115 | 0.7500 |
| FlexMatch + USE | 0.6224 | 0.6188 | 0.6228 | 0.6214 | 0.6240 | 0.6219 | 0.0029 | 0.0071 | 0.0200 | -0.0180 | 0.5000 |
| UDA | 0.6213 | 0.6181 | 0.6186 | 0.6133 | 0.6160 | 0.6175 | -0.0077 | 0.0112 | 0.0135 | -0.0265 | 0.5000 |
| UDA + USE | 0.6287 | 0.6231 | 0.6190 | 0.6233 | 0.6216 | 0.6231 | -0.0070 | 0.0114 | 0.0215 | -0.0280 | 0.2500 |

**General performance.** Compared with the CIFAR-100 experiments, the absolute performance is lower (around 0.60–0.62), reflecting the difficulty of text classification under low-label conditions.

**Effect of USE.** The role of USE in NLP is more nuanced. In contrast to CIFAR-100 with 200 labels, where USE was essential to prevent collapse, here the baseline models are already stable. USE mainly serves as a light-weight stabilizer: it reduces GM and BAD values in several cases, flattens slopes, and slightly improves adjacent-drop probabilities $P$. For example, FixMatch under far-OOD improves its GM from $0.0105$ to $0.0305$ with USE, while FlexMatch maintains nearly constant curves across contamination levels. However, the magnitude of gains is smaller than in vision, underscoring that textual distributions are less prone to catastrophic OOD drift.

## A.4 USE OF LARGE LANGUAGE MODELS (LLMS)

We used a large language model to assist with language editing. Specifically, the model was used to transform draft text fragments that we had written into more readable and fluent sentences. The model was not used for generating research ideas, designing experiments, analyzing results, or draw-

ing conclusions. All substantive content and scientific contributions in this paper are solely the work of the authors.

