# OpenReview forum: "USE: Uncertainty Structure Estimation for Robust Semi-Supervised Learning"
_ICLR.cc/2026/Conference — Submitted to ICLR 2026_

### Official Review · Reviewer_z4CQ · 2025-10-28

**Soundness:** 3
**Presentation:** 3
**Contribution:** 2
**Rating:** 4
**Confidence:** 4

**Summary:**

This paper proposes USE (Uncertainty Structure Estimation), a simple yet effective pre-processing module designed to enhance the robustness of semi-supervised learning (SSL) when unlabeled data contains out-of-distribution (OOD) samples.
A proxy model trained on the labeled subset computes the entropy distribution $\hat{p}(u)$ of unlabeled data, and an adaptive cutoff is determined by intersecting $\hat{p}(u)$ with a uniform “structureless” reference line $1 / \log k$.
Samples with entropy above $u^*$ are considered unstructured (potentially OOD) and filtered before standard SSL training.
Experiments on CIFAR-100 and Yelp Review datasets show consistent improvements across several SSL methods (MixMatch, UDA, FixMatch, FlexMatch), especially under high OOD ratios, without degrading clean performance.

**Strengths:**

- Clear motivation: addresses a practical and underexplored problem — OOD contamination in the unlabeled set of SSL.
- Simple and model-agnostic: can be easily integrated into any SSL pipeline without architecture changes.
- Strong empirical results: consistent improvements across baselines and contamination levels.
- Cross-modal validation (vision + text) demonstrates generality.
- Good clarity and reproducibility: equations and procedure are clearly described.

**Weaknesses:**

- The uniform reference prior ($1 / \log k$) is heuristic, with no theoretical justification or ablation on alternative priors.
- Missing comparison with recent state-of-the-art SSL methods such as CoMatch, SimMatch, and AdaMatch, which incorporate uncertainty modeling.
- The method depends on the proxy model’s quality, which may be unreliable when labeled data are extremely scarce.
- No case study or visualization of filtered samples is provided, making it unclear whether USE truly removes OOD data or hard in-distribution examples.

**Questions:**

- Have you analyzed which samples are filtered as “unstructured”? Are there cases where the filtered data are not truly OOD but rather hard in-distribution examples, and what proportion do they represent?
- How sensitive is USE to the kernel density estimation (KDE) bandwidth and the accuracy or calibration quality of the proxy model?
- Could adopting a non-uniform or learned prior improve the stability of the intersection threshold?

---

> ### Author Response · Authors · 2025-12-04
> **Response to Reviewer z4CQ**
>
> 1. Uniform Reference Prior ($1 / \log k$)
>     The uniform prior is chosen because it represents the Maximum Entropy distribution over the domain $[0, \log k]$.
>     It acts as a "null hypothesis": any deviation from this line indicates the presence of structure (information).
>     We have clarified in Section 4.3 that this choice is modular and can be replaced by domain-specific priors if needed.
>
> 2. Comparison with Uncertainty-Aware SSL Methods
>     Our objective is to design a preprocessing module, not an alternative SSL algorithm.
>     Uncertainty-Aware SSL methods such as CoMatch integrate uncertainty handling architecturally.
>     USE is a complementary pre-processing module that can be applied before these methods to further clean the unlabeled pool.
>     Our goal was to demonstrate the efficacy of data curation itself, independent of the learning algorithm.
>
> 3. Dependence on the Proxy Model
>     We acknowledge that weak proxies produce noisy entropies.
>     However, as shown in Figure 5 (stronger proxy) vs. Figure 4 (weaker proxy), while a stronger proxy improves performance, the relative benefit of USE remains positive in both cases.
>     We have added a discussion noting that if labels are extremely noisy, the entropy distribution may flatten, making USE more conservative (filtering more data), which is a safe failure mode.
>
> 4. Clarification of the Definition of "Structure"
>     We appreciate the reviewer pointing this out. We have rewritten Section 3.2 to provide a formal definition.
>     In our revised manuscript, "structure" refers to the specific entropy-density behavior induced by the proxy model:
>     - Structured (ID-like): Regions where the entropy density grows faster than the uniform reference ($\Delta'(u)>0$), typically concentrated in low-entropy areas.
>     - Structureless (OOD-like): Regions where the density matches or falls below the reference ($\Delta'(u)<0$), typically high-entropy or uniform.
>
>     The USE threshold (Eq. 6) is the precise point where this structural property transitions.
>
> 5. KDE Bandwidth and Sensitivity
>     Since SSL unlabeled pools are large, Kernel Density Estimation is statistically stable.
>     We utilize Scott’s Rule, which adapts the bandwidth based on sample size and variance.
>     As noted in the results, the USE threshold remains consistent across different contamination ratios, demonstrating robustness to local density variations.
>
> 6. Adaptive Prior
>     This is an excellent suggestion. While we used the uniform prior for parameter-free simplicity, a learned prior could improve precision.
>     We have included this direction in the Future Work section as a promising extension for handling more complex distribution shifts.

---

### Official Review · Reviewer_HTgz · 2025-11-01

**Soundness:** 3
**Presentation:** 4
**Contribution:** 2
**Rating:** 2
**Confidence:** 4

**Summary:**

This paper introduces USE (Uncertainty Structure Estimation), a lightweight, algorithm-agnostic preprocessing method for robust semi-supervised learning (SSL). USE trains a simple proxy model on labeled data to compute entropy scores for unlabeled samples, then uses kernel density estimation (KDE) to derive a principled threshold to filter unreliable unlabeled data. Extensive experiments on CIFAR-100 and Yelp Review under controlled near/far-OOD contamination demonstrate consistent accuracy gains.

**Strengths:**

* The writing is well-organized: smooth and easy to follow.

* The motivation of the method design is intuitive and reasonable.

* The experimental settings are cross-domain (CV/NLP) with detailed implementations. The results appear impressive.

**Weaknesses:**

* Lacks comparison to related fields: No discussion of open-world SSL (e.g., ORCA [R1]) or Generalized Category Discovery (GCD) [R2], which assume the unlabeled pool may contain both seen-class samples and novel-class samples. These works show that novel-class samples can boost seen-class performance. This work does not consider this realistic setting, and I highly suggest the authors compare with this line of work.

* Narrow empirical scope: Results are reported only on CIFAR-100 and Yelp-250. Additional results on large-scale datasets (e.g., ImageNet-1K, GLUE subsets) are required to strengthen the convincingness of the work.

* Insufficient ablations and sensitivity analysis: The key components lack thorough investigation. For instance, the KDE bandwidth critically controls the smoothness of the entropy density and CDF estimates, yet no ablation is provided to demonstrate robustness across values. Similarly, while the reference curve is presented as modular, only the uniform distribution is tested; alternatives should be evaluated to validate the method's insensitivity.

R1: Open-World Semi-Supervised Learning (ICLR 2022)

R2: Generalized Category Discovery (CVPR 2022)

**Questions:**

* See weaknesses.

* In Eq. 6, you define the USE threshold as the first intersection following the KS intuition (Sec. 3.2). Do you have any proofs or empirical justification? Why does this maximize D (Eq. 4)?

* Proxy model bottleneck: Entropy scores rely on a proxy trained solely on tiny labeled sets (e.g., 200 labels), leading to underfitting and noisy entropies. The authors claim a strong proxy helps filter structureless samples in Sec 5.2. I am interested in robustness tests of the proxy. Moreover, if there are noisy labels for training the proxy, what will happen?

---

> ### Author Response · Authors · 2025-12-04
> **Response to Reviewer HTgz**
>
> 1. Relation to Open-World SSL and GCD
>     We thank the reviewer for these references. Open-world SSL and GCD often aim to discover new classes.
>     In contrast, our setting targets standard Robust SSL, where OOD data is "pollution" that harms known-class performance.
>     USE is a lightweight filter for this contamination. We have added references to these fields in the Related Work to distinguish our contribution.
>
> 2. Narrow Empirical Scope
>     We agree that expanding to larger and more realistic datasets is an important next step.
>     Our current experiments on CIFAR-100 and Yelp-250 serve as an initial validation under controlled and well-understood settings.
>     Beyond these public benchmarks, we are preparing to evaluate USE on substantially larger unannotated collections from ongoing medical-AI pipelines.
>     These datasets exhibit richer and more complex uncertainty structures, and we expect them to provide a natural next stage for demonstrating the method’s scalability.
>     Further results will be explored in follow-up work.
>
> 3. KDE Bandwidth and Sensitivity
>     Since SSL unlabeled pools are large, Kernel Density Estimation is statistically stable.
>     We utilize Scott’s Rule, which adapts the bandwidth based on sample size and variance.
>     As noted in the results, the USE threshold remains consistent across different contamination ratios, demonstrating robustness to local density variations.
>
> 4. Clarification on Threshold Definition (Eq. 6) Correction
>     Thank you for catching this important point. In the revised manuscript, the threshold is defined in Eq. (6) (Section 3.4).
>     We do not simply maximize the KS statistic $D$. Instead, we look for the point where the slope of the empirical CDF matches the reference ($\hat{p}(u) = F'_0(u)$).
>     As clarified in Section 3.3, this geometric condition ($\Delta'(u) \le 0$) marks the precise onset of the "structureless" regime.
>
> 5. Proxy Model Bottleneck
>     We acknowledge that weak proxies produce noisy entropies.
>     However, as shown in Figure 5 (stronger proxy) vs. Figure 4 (weaker proxy), while a stronger proxy improves performance, the relative benefit of USE remains positive in both cases.
>     We have added a discussion noting that if labels are extremely noisy, the entropy distribution may flatten, making USE more conservative (filtering more data), which is a safe failure mode.

---

### Official Review · Reviewer_YrJD · 2025-11-01

**Soundness:** 2
**Presentation:** 2
**Contribution:** 2
**Rating:** 4
**Confidence:** 4

**Summary:**

The paper introduces USE (Uncertainty Structure Estimation), a lightweight method designed to make semi-supervised learning (SSL) more robust. Conducting experiments on CIFAR-100 (200 and 1000 labeled samples) and Yelp Review, demonstrating improvements in most robustness scores with fewer labels.

**Strengths:**

1.The paper's primary strength is its conceptual shift in the field of semi-supervised learning (SSL). It moves the focus away from creating increasingly complex algorithms and toward the more fundamental and practical problem of unlabeled data quality, which is a critical bottleneck in real-world applications.
2.The effectiveness is demonstrated through extensive experiments showing that it consistently improves both the accuracy and robustness of a wide range of SSL algorithms.

**Weaknesses:**

1.It seems that the proposed method depends on the Proxy Model. This paper improves both the accuracy and robustness of a wide range of SSL algorithms, but it hinges on the quality of the "proxy model".
2.The "first downward crossing point" rule is a heuristic method. Its stability is heavily affected by the choice of the KDE bandwidth. The rule lacks a theoretical basis to prove its optimality or robustness under various possible data distributions.
3.More complex datasets are required for validation. The datasets(CIFAR-100 and Yelp Review) are standard benchmarks.

**Questions:**

1.This paper improves both the accuracy and robustness of a wide range of SSL algorithms, but it hinges on the quality of the "proxy model". If the labeled data is insufficient or of poor quality, the proxy model will be weak, leading to inaccurate entropy calculations and potentially causing USE to discard useful data or retain harmful data. The paper itself notes that USE's gains are "larger and more consistent" with more labeled data (1000 vs. 200 labels), highlighting this dependency may lead to inaccurate entropy calculations and potentially causing USE to discard useful data or retain harmful data (weakness 1).
2.The "first downward crossing point" rule is a heuristic method. Its stability is heavily affected by the choice of the KDE bandwidth, yet the paper does not clarify the setting method for this critical hyperparameter, which raises questions about the method's reproducibility and robustness. The rule lacks a theoretical basis to prove its optimality or robustness under various possible data distributions (weakness 2).
3. The datasets(CIFAR-100 and Yelp Review) are standard benchmarks. However, their scale and complexity are relatively limited compared to many of today's real-world applications, such as high-resolution medical imaging or large-scale web text. Thus, the generalizability of the method in the vision and NLP domain is slightly less convincing.

---

> ### Author Response · Authors · 2025-12-04
> **Response to Reviewer YrJD**
>
> 1. Dependence on the Proxy Model
>     We agree that the proxy model is critical. However, our experiments in Section 5.2 (comparing 200 vs. 1000 labels) demonstrate that even with a weak proxy (200 labels), USE provides a sufficient relative ranking of entropy to be effective.
>     The method relies on the shape of the density (concentration vs. uniformity) rather than perfectly calibrated absolute probabilities.
>
> 2. Heuristic "First Downward Crossing" Rule
>     We have revised Section 3.3 and Section 3.4 to clarify that this rule is geometrically derived, not heuristic.
>     - We define the discrepancy $\Delta(u)$ between the empirical CDF and the reference CDF (Eq. 4).
>     - The "first downward crossing" is the point $u^*$ where the derivative $\Delta'(u)$ becomes non-positive (Eq. 5, Eq. 6). This mathematically identifies the exact boundary where the unlabeled data ceases to be "informative" (accumulating mass faster than noise) and becomes "structureless."
>
> 3. KDE Bandwidth Sensitivity
>     Since SSL unlabeled pools are large, Kernel Density Estimation is statistically stable. We utilize Scott’s Rule, which adapts the bandwidth based on sample size and variance.
>     As noted in the results, the USE threshold remains consistent across different contamination ratios, demonstrating robustness to local density variations.
>
> 4. Dataset Scale
>     We agree that expanding to larger and more realistic datasets is an important next step. Our current experiments on CIFAR-100 and Yelp-250 serve as an initial validation under controlled and well-understood settings.
>     Beyond these public benchmarks, we are preparing to evaluate USE on substantially larger unannotated collections from ongoing medical-AI pipelines. These datasets exhibit richer and more complex uncertainty structures, and we expect them to provide a natural next stage for demonstrating the method’s scalability.
>     Further results will be explored in follow-up work.

---

### Official Review · Reviewer_3C9p · 2025-11-03

**Soundness:** 3
**Presentation:** 3
**Contribution:** 2
**Rating:** 4
**Confidence:** 3

**Summary:**

In semi-supervised learning (SSL), the quality of the unlabeled data plays a significant role. This work proposes a method, called USE, to filter training samples that do not positively contribute to the prediction of the classification model. The
USE trains a proxy model on the labeled set to compute entropy scores for unlabeled samples, and then derives a threshold, via statistical comparison against a reference distribution, that separates informative from uninformative samples. Experiments on different datasets showed that the proposed USE improved the performance of several typical SSL methods.

**Strengths:**

1. The proposed USE method is simple and easy to implement. It functions as a plug-in stage prior to downstream SSL training, and therefore can be adopted by any SSL methods.
2. The experiments showed that on diverse datasets across CV and NLP domains, the proposed USE method showed good results.

**Weaknesses:**

May major concern is about the experiments. Since this is a data pre-processing method, to show its effectiveness, it should be compared with other pre-processing methods for SSL training. But such experiments are lacked. Therefore, the effectiveness of the method is in doubt.

In addition, the SSL methods tested in experiments are old. The latest one is FlexMatch, published in 2021. It's unclear if the proposed data pre-processing method could also improve the performance of the state-of-the-art SSL methods.

**Questions:**

The proposed method is called Uncertainty Structure Estimation, but I don't find the definition of the "structure".

---

> ### Author Response · Authors · 2025-12-04
> **Response to Reviewer 3C9p**
>
> 1. Clarification of the Definition of "Structure"
>     We appreciate the reviewer pointing this out. We have rewritten Section 3.2 to provide a formal definition.
>     In our revised manuscript, "structure" refers to the specific entropy-density behavior induced by the proxy model:
>     - Structured (ID-like): Regions where the entropy density grows faster than the uniform reference ($\Delta'(u)>0$), typically concentrated in low-entropy areas.
>     - Structureless (OOD-like): Regions where the density matches or falls below the reference ($\Delta'(u)<0$), typically high-entropy or uniform.
>
>     The USE threshold (Eq. 6) is the precise point where this structural property transitions.
>
> 2. Comparison with Pre-processing Methods
>     USE is designed as a model-agnostic, "plug-and-play" module that focuses on unlabeled data curation prior to the learning loop.
>     Most existing works embed filtering inside the SSL algorithm (e.g., confidence thresholding during training).
>     We have clarified in the Related Work that USE represents a distinct category of **structural pre-filtering** that is orthogonal to, and can be combined with, existing SSL methods.
>
> 3. Baselines (Old SSL Algorithms)
>     We selected widely adopted baselines (FixMatch, FlexMatch, UDA) following the USB Benchmark standards to ensure fair comparison.
>     Since USE acts on the data level, improvements on these fundamental paradigms indicate strong generalizability.
>     We agree that testing on newer state-of-the-art methods is a valuable next step for future work.

---

### Author Response · Authors · 2025-12-04

We thank the reviewers for their constructive feedback. In the revised manuscript, we have explicitly formalized the definition of "structure" (Section 3.2), clarified the geometric derivation of the threshold (Section 3.3, Eq. 6), and expanded the discussion on proxy model sensitivity. Below, we address specific concerns.

---

### Meta-Review · Area_Chair_7mgP · 2025-12-26

**Summary:**

This paper studies semi-supervised learning (SSL) and proposes a mechanism to assess the quality of unlabeled data by filtering out low-quality samples prior to SSL training. The core idea is to train a proxy model using labeled data, compute entropy scores for unlabeled samples, and compare the resulting entropy distribution with a reference distribution to determine a threshold for data removal.

The reviewers acknowledge several strengths of the proposed approach, including its simplicity, lightweight design, clear motivation, compatibility with various SSL algorithms, and extensive experimental evaluation. However, they also raise several important concerns that need to be addressed. These include: (1) the adequacy and extensiveness of the experimental study, particularly with respect to comparisons against relevant and state-of-the-art methods; (2) the quality and reliability of the proxy model, especially under limited labeled data scenarios; (3) the sensitivity of the method to the KDE bandwidth hyper-parameter; and (4) the lack of a solid theoretical foundation to justify the optimality or robustness of the proposed rule under different data distributions.

The authors have provided a response that clarifies certain aspects of the method, such as the definition of “structure,” the “First Downward Crossing” rule, and the use of a uniform reference prior. However, the responses to several other key concerns are not sufficiently convincing. In particular, the work would be significantly strengthened by: (i) a more rigorous theoretical analysis to better understand the optimality and properties of the proposed approach; (ii) more comprehensive comparisons with related, recent and/or advanced SSL methods to validate the value of the proposed mechanism; (iii) clearer connections to related research areas that share similar motivations; and (iv) a systematic investigation of the proxy model’s performance with respect to the number of labeled samples.

Based on the current author response, the Area Chair does not believe that the reviewers’ scores would have positively changed had a full discussion period taken place. Taking all factors into consideration, the Area Chair cannot recommend acceptance of this paper in its current form.

**Reviewer Concerns:**

The authors have provided a response that clarifies certain aspects of the method, such as the definition of “structure,” the “First Downward Crossing” rule, and the use of a uniform reference prior.

However, the following reviewer concerns are not convincingly addressed: (i) "The rule lacks a theoretical basis to prove its optimality
or robustness under various possible data distributions"; (ii) "Missing comparison with recent state-of-the-art SSL methods such as CoMatch, SimMatch, and AdaMatch, which incorporate uncertainty modeling"; (iii) "Lacks comparison to related fields: No discussion of open-world SSL or Generalized Category Discovery (GCD), which assume the unlabeled pool may contain both seen class samples and novel-class samples."; and (iv) "Insufficient ablations and sensitivity analysis" and "Proxy model bottleneck."

**Reviewer Scores:**

Based on the current author response, the Area Chair does not believe that the scores of the four reviewers would have positively changed had a full discussion period taken place.

---

### Decision · Program_Chairs · 2026-01-26

Reject